# Blocking IbmiR319a Impacts Plant Architecture and Reduces Drought Tolerance in Sweet Potato

**DOI:** 10.3390/genes13030404

**Published:** 2022-02-24

**Authors:** Lei Ren, Tingting Zhang, Haixia Wu, Xinyu Ge, Huihui Wan, Shengyong Chen, Zongyun Li, Daifu Ma, Aimin Wang

**Affiliations:** 1Institute of Integrative Plant Biology, School of Life Science, Jiangsu Normal University, Xuzhou 221116, China; r15905217129@163.com (L.R.); 15396884589@163.com (T.Z.); whx2734793896@163.com (H.W.); gxy20000703@163.com (X.G.); wanyang2003@outlook.com (H.W.); zongyunli@jsnu.edu.cn (Z.L.); 2Jiangsu Key Laboratory of Phylogenomics & Comparative Genomics, School of Life Science, Jiangsu Normal University, Xuzhou 221116, China; 3Zhanjiang Academy of Agricultural Sciences, Zhanjiang 524094, China; chengreen-1980@163.com; 4Key Laboratory for Biology and Genetic Breeding of Sweetpotato (Xuzhou), Ministry of Agriculture/Jiangsu Xuzhou Sweetpotato Research Center, Xuzhou Institute of Agricultural Sciences, Xuzhou 221131, China

**Keywords:** sweet potato, microRNA319, TCP transcription factor, drought stress

## Abstract

MicroRNA319 (miR319) plays a key role in plant growth, development, and multiple resistance by repressing the expression of targeted *TEOSINTE BRANCHED/CYCLOIDEA/PCF* (*TCP*) genes. Two members, *IbmiR319a* and *IbmiR319c*, were discovered in the miR319 gene family in sweet potato (*Ipomoea batatas* [L.] Lam). Here, we focused on the biological function and potential molecular mechanism of the response of *IbmiR319a* to drought stress in sweet potato. Blocking *IbmiR319a* in transgenic sweet potato (MIM319) resulted in a slim and tender phenotype and greater sensitivity to drought stress. Microscopic observations revealed that blocking *IbmiR319a* decreased the cell width and increased the stomatal distribution in the adaxial leaf epidermis, and also increased the intercellular space in the leaf and petiole. We also found that the lignin content was reduced, which led to increased brittleness in MIM319. Quantitative real-time PCR showed that the expression levels of key genes in the lignin biosynthesis pathway were much lower in the MIM319 lines than in the wild type. Ectopic expression of IbmiR319a-targeted genes *IbTCP11* and *IbTCP17* in *Arabidopsis* resulted in similar phenotypes to MIM319. We also showed that the expression of *IbTCP11* and *IbTCP17* was largely induced by drought stress. Transcriptome analysis indicated that cell growth-related pathways, such as plant hormonal signaling, were significantly downregulated with the blocking of *IbmiR319a*. Taken together, our findings suggest that *IbmiR319a* affects plant architecture by targeting *IbTCP11/17* to control the response to drought stress in sweet potato.

## 1. Introduction

Drought is a major environmental factor causing abiotic stress [1,2,3]. Severe drought restricts crop growth and significantly reduces yield worldwide [4,5]. It is therefore essential that crops with drought tolerance traits are produced. Generally, the responses of plants to abiotic stress are similar, especially in the first phase—a rapid osmotic phase that inhibits shoot growth [6]. At the physiological level, the symptoms of drought damage include wilting, growth retardation through reduced photosynthetic capacity (especially the lost function of PSII) [7], discoloration, abnormal ripening, and so on [8].

Although sweet potato (*I batatas*) is a drought-tolerant root crop, dry matter accumulation and root tuber enlargement are inhibited under drought stress, which seriously hampers production [9]. Genetic engineering methods can be used to improve drought tolerance. At the molecular level, the ectopic expression of enzyme genes, such as *XvSap1* (*Xerophyta viscosa* stress-associated proteins) [10], *SoBADH* (spinach betaine aldehyde dehydrogenase) [11], *AgcodA* (*Arthrobacter globiformis* choline oxidase), *AtHDG11* (*Arabidopsis thaliana* Homeodomain glabrous 11) [12], and *XvAld1* (*Xerophyta viscosa* Aldose reductase 1) [13], improves drought tolerance in sweet potato. Overexpression of endogenous enzyme genes and structural genes, such as *IbC4H* (cinnamate 4-hydroxylase) [14], *IbMIPS1* (myo-inositol-1-phosphate synthase) [15], *IbCBF3* (dehydration-responsive element-binding/C-repeat-binding factor) [16], *IbNHX2* (Na^+^/H^+^ antiporters) [17], *IbBT4* (BTB-TAZ-domain protein) [18], and *IbARF5* (auxin response factor) [19], or transcription factors, such as *IbbZIP1* (basic region/leucine zipper motif transcription factor) [20], *IbABF4* (abscisic acid (ABA)-responsive element binding factors) [21], and *IbMYB116* [22], can also improve the drought tolerance of transgenic sweet potato plants.

The above studies mainly focused their attention on maneuvering downstream gene functioning in the physiological responses of osmotic or ionic adjustment in sweet potato. The upstream regulatory networks of stress responses in sweet potato are still largely unknown. Increasingly more research has revealed that microRNAs (miRNAs) are involved in various plant stress responses by regulating their target genes, which are mainly transcription factors, forming a complex regulatory network and playing key roles in the gene regulation networks [23]. MiRNAs are small single-stranded non-protein-coding RNAs usually 20–24 nucleotides (nt) in length [24,25] that regulate gene expression by mRNA cleavage at the post-transcriptional level or translational repression through base pairing with the complementary sequence within the target mRNAs involved in plant growth, development, and also various plant stress responses [26]. MicroRNA319 (miR319) belongs to one of the most ancient and conserved miRNA families [27]. Increasingly more studies have shown that miR319 targets transcription factor *TCP* genes, playing vital roles in plant morphogenesis and reproduction [28,29,30], and also responding to multiple biotic and abiotic stresses [31,32,33,34]. In leaf development, the overexpression of miR319 or suppression of its target *TCP* genes contributed to directly regulate the progression of the cell cycle genes *ICK1*/*KRP1* to control the G_2_-M phase of the cell cycle [30], causing an uneven leaf shape and curvature and resulting in serrated leaves in *Arabidopsis* [28]. The miR319-TCP module also affects the balance between mitosis and endoreduplication [35]. In the response to various stresses, overexpressing *shamiR319d* enhanced temperature tolerance by inhibiting the expression of *GAMYB-like1* and further altering temperature and reactive oxygen species’ [36] signal transduction in tomato [37]. The overexpression of both *OsaMIR319a* and *OsaMIR319b* led to enhanced cold tolerance by downregulating *OsPCF5* and *OsPCF8* in rice [31]. *PvmiR319*, targeting *PvPCF5*, promoted ethylene (ET) synthesis to improve salt tolerance in switchgrass (*Panicum virgatum* L.) [38]. The miR319a/TCP module participated in trichome initiation synergistically with gibberellic acid (GA) signaling and improved insect defenses in *Populus tomentosa* [34].

Although miR319-mediated changes in plant morphology and the response to biotic and abiotic stresses have been well studied in many plants, there is limited information available for sweet potato, which is a widely cultivated and important tuberous crop. In this study, we explored the function of the miR319-TCP module in the plant architecture and the tolerance of drought in sweet potato. We produced transgenic sweet potato plants with inhibited *IbmiR319a* by miRNA target MIMICS (MIM319), which has been proven to be efficient in inhibiting miRNA function [39]. Blocking *IbmiR319a* not only affected plant architecture, but also reduced drought tolerance in sweet potato seedlings. Moreover, *IbTCP11/17*, two target genes of *IbmiR319a*, participated in drought stress. Collectively, the results suggest that the miR319-IbTCP module modulates plant architecture, thereby influencing drought tolerance in sweet potato.

## 2. Materials and Methods

### 2.1. Plant Materials

The sweet potato cultivar ‘Xushu 22’ wild type (WT), developed by the Sweet Potato Research Institute of the China Agriculture Academy of Science (SPRI-CAAS), was used as a donor for genetic transformation. Untransformed and transgenic plants subcultured from in vitro plantlet cultures were transferred into plastic pots (18 cm in diameter) containing dark soil and vermiculite at a ratio of 2:1 (*v*/*v*) and grown in a growth chamber under a 16 h light/8 h dark photoperiod at 25 ± 3 °C. Shoots that were 3–4 cm in height were transplanted into the field in early May in 2020 for evaluation of the phenotype and agronomic traits at the Xuzhou experimental station (E 117°17.48′, N 34°16.95′) of the Sweet Potato Research Institute of the Chinese Academy of Agricultural Sciences (SPRI-CAAS).

The *Arabidopsis thaliana* plants were grown in an artificial climate chamber at 22 ± 3 °C and 16 h light/8 h dark photoperiod.

### 2.2. Plasmid and Sweet Potato Genetic Transformation

p35S-MIM319, the miR319 target mimicry vector, was constructed as previously described [39]. Genetic transformation of sweet potato was conducted according to the previously described method by Yang [40].

### 2.3. Vector Construction and Arabidopsis Transformation

The full-length CDSs of the *IbTCP11/17* sequences were amplified using gene-specific primers, and then cloned into the S*ac*I and K*pn*I sites of the binary vector pCAMBIA1300 containing the cauliflower mosaic virus 35S promoter to create the overexpression vectors IbTCP11OE and IbTCP17OE, respectively, and then transferred into *Agrobacterium tumefaciens* strain LBA4404. *Arabidopsis* transformation was performed using the floral-dip method to produce transgenic *Arabidopsis* plants [41], which were subsequently grown in pots to produce T_3_ seeds by screening with 50 mg/L hygromycin.

### 2.4. RNA Isolation and qRT–PCR Analysis

Total RNAs were extracted using TRIzol (Invitrogen, Carlsbad, CA, USA) and then treated with RNase-free DNase I (Sigma, St. Louis, MO, USA) to remove contaminated genomic DNA. The RNA integrity was determined by 1% gel electrophoresis, and the RNA concentration was measured using a NanoDrop spectrophotometer (ND1000, Technologies, Wilmington, NC, USA).

For quantitative real-time PCR (qRT-PCR) analyses of mRNA for the genes, 2 μg of RNA per sample were reverse-transcribed to produce cDNA using the HiScript II 1st Strand cDNA Synthesis Kit (Vazyme, Shanghai, China). The qRT-PCR was performed with the ClonExpress Ultra One Step Cloning Kit (Vazyme), and the *IbActin* gene was used as an internal control. Data from three biological samples were collected, and the mean values were normalized to *IbActin*.

For miRNA quantitative analysis, the stem-loop RT-PCR method was used. One microgram of DNase-treated RNA was converted into cDNA using the First Strand cDNA Synthesis Kit (Takara, Dalian, China). The miR319-specific primer and miRNA universal primer (URP) were used for qRT-PCR. *IbActin* RNA was used as an endogenous control. The abundance of *miR319* was normalized to *IbActin* RNA as a reference.

The relative abundance of gene expression was determined using the 2^−ΔΔCt^ method [42]. All gene expression data are from three biological replicates with three technical replicates for each biological sample. The sequences for the primers are listed in Appendix A.

### 2.5. Microscopic Observations

For paraffin sectioning, the first fully expanded fresh leaves or stems were fixed in 0.1 M sodium phosphate buffer containing 2.5% (*v*/*v*) glutaraldehyde for 24 h. Samples were dehydrated through an alcohol series, followed by resin-alcohol grading, and embedding in acrylic resin. Semi-thin sections of 1 mm thickness were obtained using the standard rotary microtomy technique and stained with Safranin O-Fast Green. Photomicrographs were taken under 10× and 20× objectives of the fluorescence microscope (Axiolab, Zeiss, MC80 Dx Camera).

For epidermis cell observation, the first fully expanded fresh leaves were folded and torn gently from the fold, and a small piece of torn white film was made into a temporary slide and observed with a light microscope. The length and width of the epidermis cells were measured, and the stoma were counted.

### 2.6. Lignin Deposition Experiment and Lignin Content Measurement

Analysis of the Klason lignin content was performed with the sulfuric acid digestion method [43]. For lignin deposition staining with toluidine blue, hand-cut sections of the third internode were cut from two-month-old plants in the greenhouse. Free-hand slices were made and stained with toluidine blue (toluidine blue staining solution, Servicebio, Wuhan, China) for about 2 min, rinsed with water, and then observed under an optical microscope.

### 2.7. Measurement of Breaking Force

The third internodes from the top of two-month-old MIM319 and WT seedlings were used for measurements. The force required to break the stems was recorded with a texture analyzer (Shimadzu, EZ Test, Kyoto, Japan) with the detector TA52. Ten plants of each of MIM319 and WT were measured, and all measurements were taken under the same conditions.

### 2.8. Analysis of Drought Tolerance

The cuttings of MIM319 and WT were used for the drought stress experiments. The cuttings transplanted for 2 weeks were watered sufficiently for 1 week, following which water was withheld for 4 weeks to simulate drought stress. The physiological indexes, including malondialdehyde (MDA) and proline contents, and antioxidant enzyme system activity, were measured with assay kits (Nanjing Jiancheng Bioengineering Institute, Nanjing, China) as described before [14]. Each data point was the average of three replicates. Approximately 20 cuttings of each line were used for each experiment, and at least 3 replicates of each experiment were performed, and the results were consistent. The result from one set of experiments is presented herein.

### 2.9. Gene Expression Analysis

The first fully expanded leaves of MIM319 and WT plants subjected to drought stress for one week were used to analyze the relative expression level of the genes related to stress responses, including target genes *IbTCP11/17*, proline biosynthesis, and the ROS-scavenging system, using qRT-PCR. The sequences of the specific primers are listed in Appendix A.

### 2.10. Transcriptome Analysis

Total RNA was extracted from the shoots of MIM319 and WT grown in a transplanting box for 4 weeks using the TRIzol method (Invitrogen, Carlsbad, CA, USA), and l g of RNA was used for Illumina RNA-Seq. Transcriptome sequencing and de novo transcriptome assembly and evaluation were performed by Huada (Beijing, China). After filtering with SOAPnuke (v1.5.2). (Available online: https://github.com/BGI-flexlab/SOAPnuke, accessed on 5 August 2021), clean reads were obtained and stored in FASTQ format. The clean reads were mapped to the reference genome using HISAT2 (v2.0.4) (Available online: http://www.ccb.jhu.edu/software/hisat/index.shtml, accessed on 5 August 2021). Bowtie2 (v2.2.5) (Available online: http://bowtiebio.sourceforge.net/%20Bowtie2%20/index.shtml, accessed on 5 August 2021) was used to align the clean reads to the reference coding gene set, and then the expression level of the genes was calculated by RSEM (v1.2.12) (Available online: https://github.com/deweylab/RSEM, accessed on 5 August 2021). A heatmap was produced using pheatmap (v1.0.8) (Available online: https://cran.r-project.org/web/packages/pheatmap/index.html, accessed on 5 August 2021) according to the gene expression in different samples. The differential expression analysis was performed using DESeq2 (v1.4.5) (Available online: http://www.bioconductor.org/packages/release/bioc/html/DESeq2.html, accessed on 5 August 2021) with Q-values ≤ 0.05. Gene Ontology (GO; Available online: http://www.geneontology.org/, accessed on 5 August 2021) and Kyoto Encyclopedia of Genes and Genomes (KEGG; Available online: https://www.kegg.jp/, accessed on 5 August 2021) enrichment analyses of differentially expressed genes (DEGs) were performed by Phyper (Available online: https://en.wikipedia.org/wiki/Hypergeometric_distribution, accessed on 5 August 2021) based on the hypergeometric test. The significance levels of the terms and pathways were corrected by the Q-value with a rigorous threshold (Q value ≤ 0.05) by Bonferroni [44]. All the above website accessed on 5 November 2021.

### 2.11. Statistical Analysis

The Student’s *t*-test was used to analyze all the data presented as the mean ± SE. *p*-values of <0.01, or <0.05 were considered to be statistically significant.

## 3. Results

### 3.1. Identification and Characterization of IbmiR319 in Sweet Potato

MiR319 was the first miRNA identified through positive genetic screening [28]. In previous studies, we constructed an miRNA library of sweet potato based on high-throughput sequencing [2], and two members of the miR319 family (IbmiR319a and IbmiR319c) with different matured miRNA sequences were found (Figure 1A,B). Both IbmiR319 precursors were supported by cDNAs in the public database (http://public-genomes-ngs.molgen.mpg.de/SweetPotato/, accessed on 5 August 2021). The gene encoding *IbmiR319a* was located in chromosome (chr) 15 from 1889,2867 to 1889,3043 (Appendix A), while that of *IbmiR319c* was located in chr 2 from 4761,9325 to 4761,9514 (Appendix A).

To determine the expression patterns of *IbmiR319*, we conducted stem-loop qRT-PCR of *IbmiR319* mature transcripts in the leaves, stems, fibrous roots, pencil roots, and storage roots (Figure 1C,D). *IbmiR319a* showed a higher expression level in the young organs or vigorously growing organs, especially the young leaf and developing root, while *IbmiR319c* showed a higher expression level in the fibrous roots, indicating that *IbmiR319a* may play an important role in development. We therefore focused our attention on the functional analysis of *IbmiR319a*.

### 3.2. Generation of Transgenic Sweet Potato Plants with miR319a Blocked

To investigate the function of *IbmiR319a* in sweet potato, we constructed a plasmid overexpressing *IbmiR319a* target mimicry (MIM319), intending to sequester the normal expression of native *IbmiR319a*, and transformed it into the sweet potato variety ‘Xu22′, which was used as WT. We obtained seven independent transgenic MIM319 plants, among which five positive plants named m3-2, m3-3, m3-8, m3-9, and m3-10 were verified (Appendix A). To determine the expression level of *IbmiR319a*, these five positive transgenic lines were analyzed using stem-loop qRT-PCR, with WT as the control. The abundance of *IbmiR319a* mature transcripts in transgenic plants was lower than that in WT (Figure 2A), suggesting that the MIM319 fusion plasmids were successfully expressed in sweet potato, and normal *IbmiR319a* was successfully blocked. One independent transgenic line m3-9, with relatively lower *IbmiR319a* expression levels, was chosen as MIM319 for further analysis.

### 3.3. Expression Level of the Putative IbmiR319a Target Genes Was Upregulated in MIM319 Transgenic Sweet Potato

In general, miRNAs play important roles by post-transcriptionally regulating their target genes. In sweet potato, *IbmiR319a* was predicted to target Transcript comp94376_c4 and comp87184_c3 according to our degradation and transcriptome sequencing results [2]. Using these sequences as a reference in the BLAST search in the genomics database for its 2 wild ancestors (*Ipomoea trifida* and *Ipomoea triloba*) (Available online: http://sweetpotato.uga.edu/. accessed on 5 August 2021), we found that they were 98% identical to sequence ID CP025644.1 on chr 1 and CP025653.1 on chr 10 of *I. trifida* (Appendix A). We named them *IbTCP11* and *IbTCP17* according to their chromosome location [45]. This complementary area between the targets and *IbmiR319a* was shown in our previous study through the psRNA Target tool [45]. We further checked the predicted *IbTCP11* and *IbTCP17* transcript levels in MIM319 plants by qRT-PCR. The result showed that *IbTCP11/17* increased the mRNA levels in the leaves of transgenic MIM319 plants compared to the WT (Figure 2B,C). Considering that MIM319 plants showed a relatively low level of mature *IbmiR319a* expression in the leaves where the miR319-targeted *IbTCP11*/*17* had a high level of expression, we assumed that blocking *IbmiR319a* could promote the high mRNA levels of the targeted genes.

### 3.4. Blocking IbmiR319a in Sweet Potato Caused Pleiotropic Phenotype Changes

As previously reported, the highly conserved ancient miR319 plays an important role in plant development [31]. Moreover, overexpressing or blocking *miR319* resulted in pleiotropic phenotype changes. All positive MIM319 transgenic sweet potatoes blocking *IbmiR319a* showed similar phenotypes. Despite the narrow and small leaves previously reported [45], a more thorough investigation was conducted. Microscopic analysis of the leaf samples showed that the epidermal cells of MIM319 were shaped like irregular squares while the WT cells were irregular rectangles (Figure 2D,E). The average length of the adaxial and adaxial leaf epidermis cells was 45 and 42.22 μm in MIM319, respectively, which was not obviously different from that of the WT (43.33 and 45.56 μm, respectively). However, the average width of the abaxial and adaxial leaf epidermis cells was 19.11 and 22.56 μm in MIM319, which was significantly different from that of WT at 26.11 μm (Figure 2F,G). Meanwhile, the number of stoma was increased, especially that of the adaxial leaf epidermis, which was 84 per microscopic view in MIM319, indicating a significant increase compared with the WT (30 per microscopic view) (Figure 2H). A looser arrangement of mesophyll cells (including spongy tissue and palisade tissue) in the leaf transverse section (Appendix A) and larger intercellular spaces in the petiole transverse section were found compared to that of the WT (Appendix A).

From the whole plant level, the height of MIM319 was significantly higher than that of WT in the greenhouse (Figure 3A). After transplantation into an incubator for 1 month, the transgenic plants MIM319 were 14.48 cm tall, whereas the WT was only 9.07 cm (Figure 3B). Blocking *IbmiR319a* also led to a decreased stem diameter in the transgenic plant MIM319. The diameter of the basal stem of the transgenic plant MIM319 was 1.987 mm, which was much slenderer than that of WT at 3.62 mm (Figure 3C).

During planting, we found that the stems of MIM319 were more brittle and broke easily. We further determined the lignin content, deposition, and the load capacity of the stems. Under the same grinding conditions, the stem powder of MIM319 was finer than that of WT (Figure 3D,E). Furthermore, the total Klason lignin content of MIM319 was 296.10 μg/gFW, which was decreased compared to the 346.45 μg/gFW in WT (Figure 3F). Toluidine blue staining for lignin in the third internodes also revealed a lower number of lignified cells around the xylem, as indicated by the decreased coloration in comparison with the WT (Appendix A). Less purplish-red staining by saffron was observed in the vertical section of MIM319 stems than that in WT (Appendix A). The breaking force of MIM319 was 1938 g/mm^2^, which was also significantly less than 4226.67 g/mm^2^ in WT (Figure 3G).

All these findings suggest that blocking *IbmiR319a*, with the resulting upregulation of *IbTCP11/17*, caused alterations in the growth and development and lignin content in sweet potato.

### 3.5. RNA-Seq Analysis of Transgenic Sweet Potato Plants with Blocked IbmiR319a

To explore the potential molecular mechanisms affecting the growth and development in MIM319 plants, RNA-Seq analysis was performed using the shoots of the MIM319 lines and WT. The DEGs were identified based on adjusted *p*-values < 0.05. To validate the RNA-Seq data, the expression of 10 randomly selected DEGs was examined by qRT-PCR and was found to be consistent with that determined by RNA-Seq (Appendix A). The RNA-Seq results identified 5587 DEGs between MIM319 and WT plants, including 3119 upregulated and 2468 downregulated genes (Figure 4A, Appendix A). The GO functional annotation of the DEGs revealed that blocking *IbmiR319a* affected multiple biological processes, including the developmental process, growth, metabolic process, and signal transduction (Figure 4B, Appendix A). The KEGG pathway analysis showed that 116 DEGs were enriched in MAPK signaling transduction, and 80 DEGs were enriched in phenylpropanoid biosynthesis (Figure 4C, Appendix A). For example, the transcript CL5843.Contig1_Ib, which is homologous to longifolia1-like, functions in regulating leaf morphology by promoting cell expansion in the leaf-length direction [46,47]. The mutant of transcript CL431.Contig8_Ib, which is homologous to E3 ubiquitin-protein ligase DIS1, was defective in trichome cell expansion and actin organization, resulting in a distorted trichome phenotype [48] in *Arabidopsis.* Transcript CL11749.Contig1_Ib, defined as a SAUR family protein, plays a central role in auxin-induced acid growth [49,50].

### 3.6. Transgenic Arabidopsis Plants Overexpressing IbTCP11 and IbTCP17 Also Had a Narrow and Small Leaf Phenotype and Decreased Lignin Content

To further examine the function of *IbmiR319a*, a biological function analysis of *IbTCP11* and *IbTCP17* was conducted. We individually overexpressed each of these two genes in *Arabidopsis*. Transgenic *Arabidopsis* plants overexpressing *IbTCP11* (IbTCP11OE) and *IbTCP17* (IbTCP17OE) had narrow and small leaves, a decreased number of rosette leaves (Figure 5A,B), and a decreased lignin content (Figure 5C), and also resembled the MIM319 phenotype in sweet potato, in which *IbTCP11/17* was upregulated because of native *IbmiR319a* being blocked. The expression profile of these two target genes in various tissues was examined by qRT-PCR in our previous study [45]. *IbTCP11/17* was highly expressed in the aboveground tissues, including the shoot bud, leaf, and stem, but weakly expressed in the belowground organs. The expression level of *IbTCP11/17* was relatively negatively correlated with the expression of *IbmiR319a*, at least in the shoot buds and leaves. These results suggest that these two target genes *IbTCP11/17* might be involved in the regulation of plant architecture and lignin content.

### 3.7. Blocking IbmiR319a in Sweet Potato Resulted in Decreased Drought Tolerance

We wanted to assess whether blocking *IbmiR319a* would affect the performance of sweet potato under drought conditions due to morphological changes in the stems and leaves. MIM319 and WT plants with 3–4 mature leaves were subjected to drought for 4 weeks. All plants showed wilting, yellowing, and necrosis, although the leaves of MIM319 showed more severe withering than those of WT. After re-watering for two weeks, most WT plants survived while the MIM319 transgenic plants perished (Figure 6A).

The drought tolerance of the plants was further evaluated. No differences in the MDA contents between the MIM319 transgenic plants and WT were observed under control conditions (room temperature or normal watering). However, the MDA content of MIM319 increased by 2.65 and 2.9 times, respectively, under drought stress compared to normal conditions, whereas an increase of only 1.97 times was detected in WT (Figure 6B).

An opposite pattern was observed for the proline content. There was no difference in the proline contents between the MIM319 transgenic plants and WT under control conditions. By contrast, under drought treatment, the proline content of MIM319 was significantly increased by 1.77 and 1.94 times, respectively, which were all lower than the 2.3 times increase detected in WT (Figure 6B).

Quantitative RT-PCR analysis was also used to detect the expression level of the drought stress-responsive genes. Well-known stress-responsive genes, including a proline biosynthesis-related gene encoding a pyrroline-5-carboxylatesynthase (*IbP5CS*) and pyrroline-5-carboxylate reductase (*IbP5CR*); an abscisic acid (ABA) biosynthesis-related gene encoding a zeaxanthin epoxidase (*IbZEP*); 9-*cis*-epoxycarotenoid dioxygenase (*IbNCED*); a late embryogenesis abundant protein (*IbLEA*); a photosynthesis-related gene encoding a phosphoribulokinase (*IbPRK*); and ROS scavenging-related genes encoding superoxide dismutase (*IbSOD*), peroxidase (*IbPOD*), catalase (*IbCAT*), ascorbate peroxidase (*IbAPX*), and glutathione peroxidase (*IbGPX*), were significantly downregulated in the MIM319 transgenic plants compared to WT after two weeks of drought stress (Figure 6C,D).

These results implied that blocking *IbmiR319a* inhibited the antioxidant system, which includes the inhibited transcriptional expression and the activity of the ROS-scavenging enzymes under drought stress.

### 3.8. IbmiR319a Functions in Plant Tolerance of Drought Stress Possibly by Inhibiting IbTCP11/17 Expression in Sweet Potato

The expression level of *IbTCP11/17* in the WT sweet potato was more strongly induced by polyethylene glycol (PEG) 6000, which was used to simulate drought stress. The expression of *IbTCP11* peaked (2 times) at 12 h while that of *IbTCP17* peaked (6 times) at 6 h (Figure 7A,B). These results implied that *IbTCP11/17* might be involved in drought tolerance in sweet potato. To further confirm this, the expression level of *IbTCP11/17* responding to water withdrawal was also detected. The expression of *IbTCP11* peaked at 4 days while that of *IbTCP17* peaked at 7 days after water withdrawal (Figure 7C,D). These results confirmed that the target genes *IbTCP11/17* were indeed involved in drought tolerance in sweet potato.

## 4. Discussion

Although sweet potato is a root crop that is highly important for food security [51], research on sweet potato has lagged behind other crops, such as rice, wheat, and maize, because of the complexity of the genome and the inefficiency of genetic transformation. So far, the only functionally characterized miRNAs in sweet potato are *IbmiR2111* [52], *IbmiR828* [53], and *IbmiR408* [54]. In the present study, we found that *IbmiR319a* negatively regulated the abundance of *IbTCP11/17* mRNA, which affected plant architecture and drought tolerance in sweet potato.

In sweet potato, the miR319 family has two members that are each encoded by *IbMIR319a* and *IbMIR319c*. A growing number of studies have reported that the function of miR319 is far ranging and conserved among different species [6,33,34,37,55]. The overexpression of *miR319* or downregulation of its target genes leads to a broader and crinkled leaf phenotype in transgenic dicotyledonous plants, such as *Arabidopsis* [28], and tomato [37], but only a broader leaf phenotype in transgenic monocotyledonous plants, such as rice [31] and switchgrass [56]. In our study, blocking *IbmiR319a* led to a narrow and small leaf phenotype in transgenic sweet potato, which was the opposite phenotype to the overexpression transgenic plant. Blocking *IbmiR319a* also led to a decreased stem diameter in transgenic plants. These results suggest that the function of miR319 in leaf morphogenesis and plant growth is highly conversed in sweet potato. In addition, we found that ~5000 genes were differentially expressed in MIM319 (Figure 4), which suggested that IbmiR319 plays a global regulatory role in the multiple biological processes of sweet potato.

*AtTCP2*, an orthologous gene of *IbTCP11*, interacts with *AtCRY1* to inhibit hypocotyl elongation under blue light [57]. The function of *CsnTCP2* was shown to be suppressed by *CsnmiR319c* in the bud dormancy-activity cycle in tea plant [33]. *AtTCP2* also positively regulated the expression of *Circadian clock associated 1* (*CCA1*) and *EARLY FLOWERING 3* (*ELF3*) to affect leaf morphogenesis and flowering time and mediate the jasmonic acid (JA) signaling pathway to inhibit hypocotyl elongation [58]. *AtTCP4*, an orthologous gene of *IbTCP17*, directly activates *VND7* to participate in secondary cell wall biosynthesis and programmed cell death [59], and also directly activates *TRICHOMELESS1* (*TCL1*) and *TCL2* to suppress trichome initiation [60]. *AtTCP4* and *PIF3* antagonistically participate in photomorphogenesis and facilitate light-induced cotyledon opening in *Arabidopsis* [61]. Among the 18 *IbTCP* genes in sweet potato, there are only 4 complementarily targeted by *IbmiR319* [45]. Overexpression of the target genes *IbTCP11/17* in *Arabidopsis* (IbTCP11OE, IbTCP17OE) led to a narrow and small leaf phenotype and a decreased lignin content, which was consistent with that of MIM319 in sweet potato.

Leaves are not only important photosynthetic plant organs but are also the interface for plant water metabolism [62]. Water loss through stomatal transpiration is one of the key determinants of drought tolerance [63]. The density and distribution of stomata directly affect plant water metabolism and drought tolerance. In our study, the leaf epidermal cells were slenderer and had more stomata in MIM319 than in WT, and the mesophyll cells were more loosely arranged, and the intercellular space was larger in MIM319 than in WT, all of which may result in increased water loss and decreased drought tolerance.

Reactive oxygen species (ROS) can cause damage to the structure and function of biomolecules, which leads to oxidative stress in plants. MDA as one of the prominent ROS, is the final decomposition product of membrane lipid peroxidation and its content can reflect the degree of plant damage by various stresses. Proline is another essential member of ROS, and its accumulation can protect plants against ROS damage. In this study, we found that the MDA content increased much more in MIM319 than WT while the proline content decreased (Figure 6B). In addition, the ROS-scavenging system genes, including *IbSOD*, *IbGPX*, *IbAPX*, *IbCAT*, and *IbPOD*, were downregulated in MIM319 (Figure 6D). These results suggest that blocking miR319 decreases proline accumulation and inactivates the ROS-scavenging system, which leads to reduced drought tolerance in transgenic MIM319.

Lignin is an important polymer of phenylpropanoid compounds and plays a vital role in biotic and abiotic stress tolerance in plants. Greater lignification improves drought resistance by increasing the water retention capacity [64,65]. The overexpression of a rice HD-Zip transcription factor *OsTF1L* [66], foxtail millet (*Setaria italica*) R2R3-MYB transcription factor *SiMYB56* [67], or key genes in the lignin biosynthesis pathway, such as the 4-coumarate-CoA ligases gene in *Gossypium hirsutum* (*Gh4CL7*) [68], cinnamyl alcohol dehydrogenase gene in *Cucumis melo* L. (*CmCAD*) [69], and caffeoyl-CoA O-methyltransferase gene in *Paeonia ostii* (*PoCCoAOMT*) [70], can all significantly enhance tolerance to drought stress in transgenic plants by regulating lignin biosynthesis. In *Arabidopsis,* miR319-targeted *AtTCP4* activated *VND7* expression to increase the lignin content in the secondary cell wall [59]. In our study, blocking *IbmiR319a* led to increased brittleness and a decreased lignin content, and as a result, reduced the drought tolerance in sweet potato.

In summary, we demonstrated that miR319 from sweet potato, targeting transcription factors *IbTCP11* and *IbTCP17*, is involved in plant architecture and drought tolerance. Blocking IbmiR319, upregulating the expression of *IbTCP11* and *IbTCP17*, caused a slim and tender phenotype and greater sensitivity to drought stress. This study provides a novel miR319 gene for impacting plant architecture drought tolerance of sweet potato.

## Figures and Tables

**Figure 1 genes-13-00404-f001:**
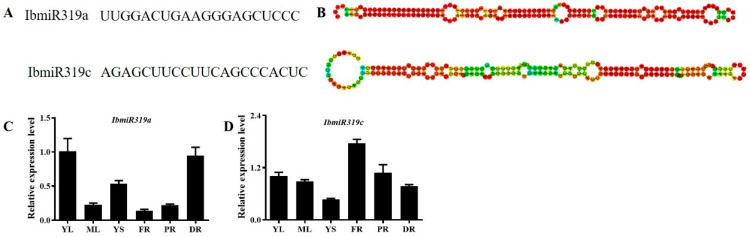
The microRNA319 family contains two members in sweet potato. (**A**): Sequence of mature *IbmiR319a* and *IbmiR319c*. (**B**): The secondary structure of the precursor sequence of *IbmiR319a* and *IbmiR319c*. (**C**,**D**): The expression patterns of *IbmiR319a* and *IbmiR319c*. YL: young leaf; ML: mature leaf; YS: young stem; FR: fibrous root; PR: pencil root; DR: developing root.

**Figure 2 genes-13-00404-f002:**
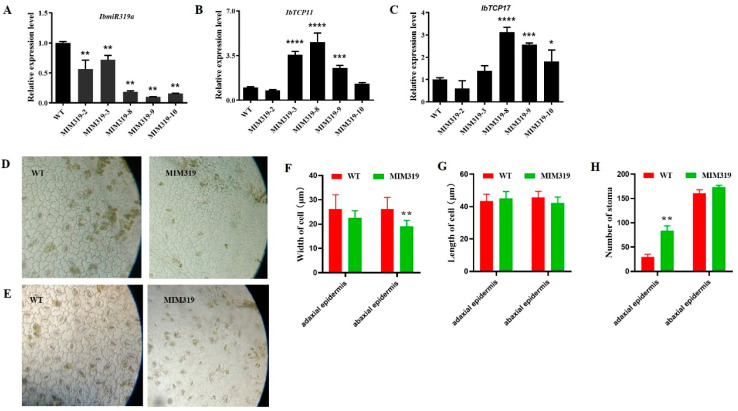
Gene expression level and light microscopic observation of MIM319 transgenic sweet potato plants. (**A**): Expression level of mature *IbmiR319a* in MIM319 transgenic sweet potato lines and WT. (**B**,**C**): Expression level of the target genes *IbTCP11/17* in MIM319 transgenic sweet potato lines and WT. (**D**): Images of leaf adaxial epidermal cells of the WT and MIM319 plants under the light microscope. (**E**): Images of leaf abaxial epidermal cells of the WT and MIM319 plants under the light microscope. (**F**): Quantitative measurement of the maximum leaf epidermis cell width of WT and MIM319 transgenic plants (*n* = 100). (**G**): Quantitative measurement of the maximum leaf epidermis cell length of WT and MIM319 lines (*n* = 100). (**H**): Statistical analysis of the total stomata number in WT and MIM319 transgenic plants (*n* = 30). Data are presented as means ± SE, and error bars represent SE. *, **, ***, **** indicate significant differences between transgenic and control plants at *p* < 0.05, 0.01, 0.001, 0.001 by Student’s *t*-test.

**Figure 3 genes-13-00404-f003:**
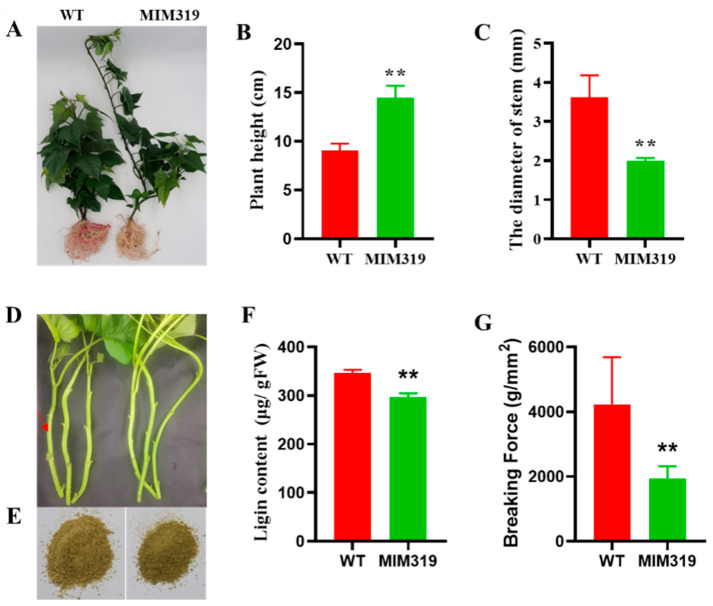
Phenotypes of MIM319 transgenic sweet potato plants. (**A**): The transgenic plants (MIM319) exhibited an increased plant height and decreased stem diameter compared to the WT controls. (**B**): Statistical analysis of plant height in MIM319 transgenic plants and WT (*n* = 20). (**C**): Statistical analysis of stem diameter in MIM319 transgenic plants and WT (*n* = 20). (**D**): A closer look at the MIM319 transgenic plants and WT. The representative transgenic plant stem is slender. The red arrow indicates the sample location. (**E**): Powdered stems of MIM319 (left) and WT (right). (**F**): Statistical analysis of lignin content in WT and MIM319 transgenic plants (*n* = 10). (**G**): The breaking force of the stems in the WT and MIM319 transgenic plants (*n* = 20). Data are presented as means ± SE, and error bars represent SE. ** indicate significant differences between transgenic and control plants at *p* < 0.01 by Student’s *t*-test.

**Figure 4 genes-13-00404-f004:**
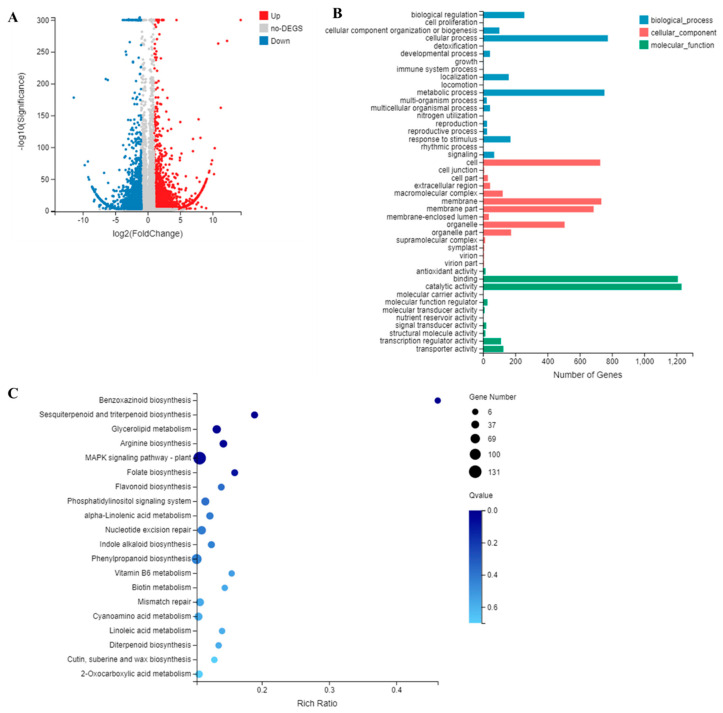
RNA-Seq analysis of MIM319 transgenic sweet potato and WT. (**A**): Volcano plot shows the differentially expressed genes (DEGs) between MIM319 transgenic sweet potato and WT. Blue circles represent downregulated DEGs; red circles represent upregulated DEGs; and grey circles represent non-DEGs. (**B**): GO analysis of the DEGs between MIM319 transgenic sweet potato and WT. (**C**): KEGG pathway analysis of the DEGs between MIM319 transgenic sweet potato and WT.

**Figure 5 genes-13-00404-f005:**
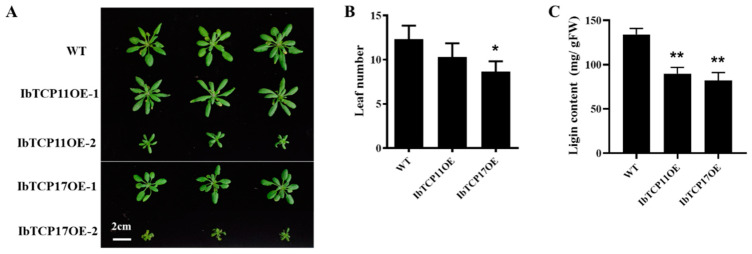
Phenotypes of transgenic *Arabidopsis* plants. (**A**): The transgenic plants (IbTCP11-OE and IbTCP17-OE) exhibited a smaller plant structure and leaf to the WT controls. Photographs of representative seedlings of WT and two transgenic lines were taken. (**B**): Leaf number in IbTCP11OE and IbTCP17OE transgenic plants and WT (*n* = 20). (**C**): Lignin content in IbTCP11-OE and IbTCP17-OE transgenic plants and WT (*n* = 10). *, ** indicate significant differences between transgenic and control plants at *p* < 0.05, 0.01.

**Figure 6 genes-13-00404-f006:**
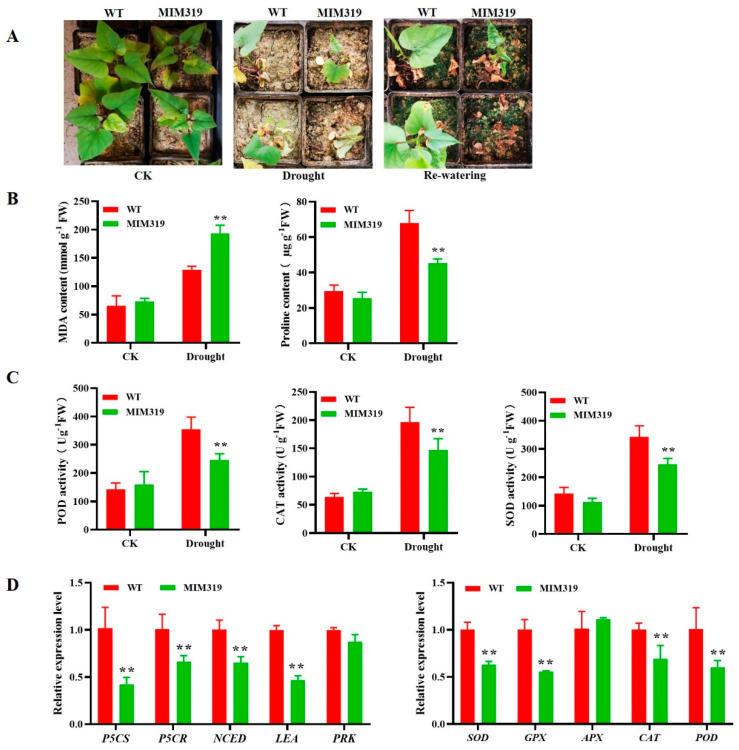
Drought tolerance analysis of MIM319 transgenic sweet potato plants. (**A**): Phenotypes of MIM319 transgenic plants vs. WT grown for 6 weeks under normal conditions (CK) and 4 weeks under drought stress (Drought) followed by 2 days of re-watering after 2 weeks of normal treatment (Re-watering). (**B**): MDA and proline contents in the MIM319 transgenic plants and WT grown for 2 weeks under drought stress after 2 weeks of normal treatment. (**C**): Peroxidase (POD), catalase (CAT), and superoxide dismutase (SOD) activity in the MIM319 transgenic plants and WT grown for 2 weeks under drought stress after 2 weeks of normal treatment. (**D**): Transcript levels of salt and drought-responsive genes in the leaves of MIM319 transgenic plants and WT plants that had been pot-grown for 4 weeks under normal conditions and 2 weeks under drought stress after 2 weeks of normal treatment. Each value is the mean of three biological repeats ± the standard deviation (SD). ** indicate significant differences between transgenic lines and WT. *p* < 0.01, Student’s *t*-test.

**Figure 7 genes-13-00404-f007:**
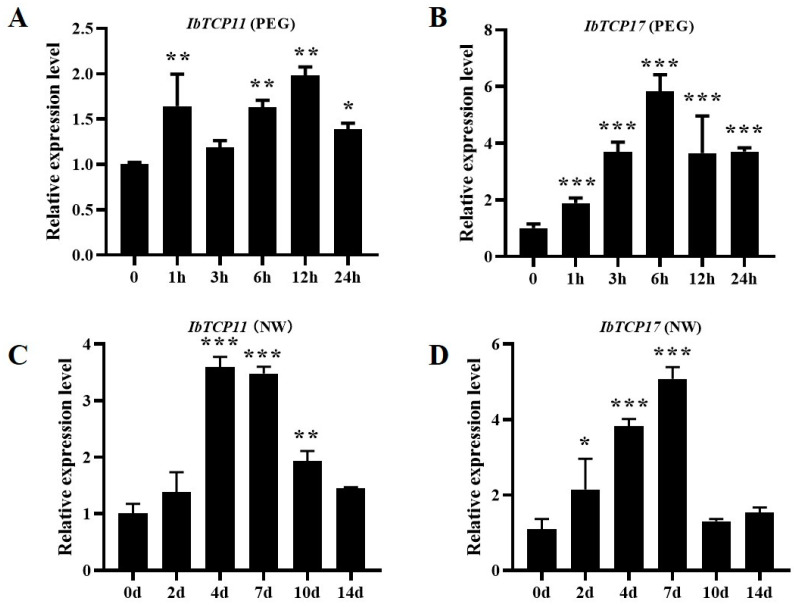
Expression analysis of the target genes *IbTCP11* and *IbTCP17* responding to drought. (**A**): Expression level of *IbTCP11* in the WT at different times (h) in response to 20% PEG. (**B**): Expression level of *IbTCP17* in the WT at different times (h) in response to 20% PEG. (**C**): Expression level of *IbTCP11* in the WT at different times (d) in response to drought stress with no water (NW). (**D**): Expression level of *IbTCP17* in the WT at different times (d) in response to drought stress with no water. Data are presented as mean values ± SE (*n* = 3). * and **, *** indicate a significant difference compared to the WT at *p* < 0.05 and <0.01, <0.001, respectively, based on Student’s *t*-test.

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
