# Peer review of "Blocking IbmiR319a Impacts Plant Architecture and Reduces Drought Tolerance in Sweet Potato"

_genes, 2022, doi:10.3390/genes13030404_

Round 1

Reviewer 1 Report

Ren et al.'s study focus on the characterization of IbmiR319 based on its effect on plant architecture and drought response. While congratulating the authors on this interesting study, I have the following comments.

Major comments

  1. The authors look at the expression of IbTCP11/17 (targets of miR319) from lines 418-424. Here authors capture the early response to PEG compared to the drought response (water withdrawal) assessed earlier. To have a meaningful comparison, the authors need to show the expression level of these two genes in the drought assay (Lines 400-417). As the authors briefly mentioned, PEG stress is more severe than water withdrawal, so it will not give a solid verification provided there are two variables, inducer, and timing. In our experience, we observed completely different responses when we tried to compare PEG, air drying, and water withdrawal. This is essential given that no temporal expression baseline is established for IbTCP11/17 under water withdrawal stress and the presence of other regulatory mechanisms of IbTCP11/17.
  2. It is interesting to see that only MDA content increases in response to drought. The possible mechanism/reason should be included in the discussion. 
  3.  The discussion needs to be further expanded. Esp. RNA-Seq results need to be further discussed.
  4. There should be a summary (concluding statement at the end). This manuscript ends very abruptly. 

Minor Comments

Reviewer 2 Report

The manuscript is well written and the presented evidence supports the conclusions being made. There are some small edits that need addressing for clarity.

Pg 1 line 45: the wording "and so on" is awkward and imprecise. Please rephrase and indicate what other processes are being referred to.

Pg 2 line 67: don't assume that everyone reading the manuscript are aware of miRNA roles in plant stress physiology.

Pg 3 Plasmid and sweet potato genetic transformation section: Include relevant protocols instead of just referring to another manuscript. Citing the original source is required but so are details needed to reproduce and evaluate the soundness of the manuscript.

The same applies to other parts of the materials and methods:

Pg 6 line 239: I don't understand the point being made here. "sequester the normal expression" is nonsensical. Please clarify and rephrase.

Round 2

Reviewer 1 Report

The authors addressed some of the suggestions made. However, Comments #1 (gene expression assay) and #3 (proper discussion on RNA-Seq data) aren't properly addressed.